# Surveillance and Control of Malaria Vectors in Hainan Province, China from 1950 to 2021: A Retrospective Review

**DOI:** 10.3390/tropicalmed8030131

**Published:** 2023-02-21

**Authors:** Dingwei Sun, Yan Chen, Lu Wang, Ximin Hu, Qun Wu, Ying Liu, Puyu Liu, Xuexia Zeng, Shangan Li, Guangze Wang, Yi Zhang

**Affiliations:** 1National Institute of Parasitic Diseases, Chinese Center for Disease Control and Prevention (Chinese Center for Tropical Diseases Research), NHC Key Laboratory of Parasite and Vector Biology, WHO Collaborating Center for Tropical Diseases, National Center for International Research on Tropical Diseases, Shanghai 200025, China; 2Hainan Provincial Center for Disease Control and Prevention, Haikou 570203, China; 3School of Global Health, Chinese Center for Tropical Diseases Research, Shanghai Jiao Tong University School of Medicine, Shanghai 200025, China

**Keywords:** review, malaria, vectors, research, Hainan Province

## Abstract

Malaria is a serious mosquito-borne tropical disease impacting populations in tropical regions across the world. Malaria was previously hyperendemic in Hainan Province. Due to large-scale anti-malarial intervention, malaria elimination in the province was achieved in 2019. This paper reviews the literature on the ecology, bionomics, and control of malaria vectors in Hainan from 1951 to 2021. We searched PubMed, and the China national knowledge infrastructure (CNKI) database for relevant articles published and included three other important books published in Chinese or English in order to summarize research on species, distribution, vectorial capacity, ecology, the resistance of malaria vectors to insecticides, and malaria vector control in Hainan Province. A total of 239 references were identified, 79 of which met the criteria for inclusion in our review. A total of six references dealt with the salivary gland infection of Anophelines, six with vectorial capacity, 41 with mosquito species and distribution, seven with seasonality, three with blood preference, four with nocturnal activity, two with flight distance, 13 with resistance to insecticides, and 14 with vector control. Only 16 published papers met the criteria of addressing malaria vectors in Hainan over the last 10 years (2012–2021). *Anopheles dirus* and *Anopheles minimus* are primary malaria vectors, mainly distributed in the southern and central areas of Hainan. Indoor residual spraying with DDT and the use of ITNs with pyrethroid insecticides were the main interventions taken for malaria control. Previous studies on ecology, bionomics, and resistance of vectors provided scientific evidence for optimizing malaria vector control and contributed to malaria elimination in Hainan Province. We hope our study will contribute to preventing malaria reestablishment caused by imported malaria in Hainan. Research on malaria vectors should be updated to provide scientific evidence for malaria vector control strategies post-elimination as the ecology, bionomics, and resistance of vectors to insecticides may change with changes in the environment.

## 1. Introduction

Malaria is a serious mosquito-borne tropical disease impacting populations in tropical regions across the world. In Africa and Southeast Asia, an estimated 247 million malaria cases were reported in 2021 [1]. With large-scale antimalarial interventions being undertaken in many regions, the World Health Organization (WHO) plans to eradicate malaria worldwide by 2050 [2]. China was officially certificated as malaria-free by the WHO in 2021.

Approximately 40 primary species of *Anopheles* are distinct in their vectorial capacity to transmit malaria vectors worldwide [2]. Local malaria transmission may be driven by primary specific vector species, which should be targeted according to their distribution, vectorial capacity, bionomics, ecology, and resistance to insecticides. Surveillance of malaria vectors is important, as it can provide accurate, granular, and timely data to aid decision-making for malaria elimination and post-elimination.

Hainan Island lies between latitude 18.10° and 20.07° N, and longitude 108.37° and 111.03° E. It is located in the south of China, and has a predominantly tropical and subtropical climate. The annual average temperature is 22–26 °C and the average rainfall is 1600 mm. The rainy and dry seasons are from May to October and November to April, respectively. The tropical environment produces an ideal region for *Anopheles* mosquitoes to breed.

Hainan occupies a land area of more than 35,400 square kilometers. Hainan is high in the middle and low towards the edges, consisting of mountains (25.4%), hills (13.3%), plateaus (32.6%), plains (28.7%), etc. With the greater socio-economic development, hill areas have been farmed, creating resting sites for adult mosquitoes, and leading to the construction of irrigation canals that are suitable for larvae [3]. Riverbeds in mountainous areas can serve as suitable breeding sites for malaria mosquitoes, especially during the rainy season [4].

Malaria was actively transmitted throughout the year in Hainan. Historically, the island was once the site of devastating malaria epidemics in China. Malaria used to be hyperendemic in Hainan Province, with more than 1000 cases per 10,000 people reported in 1951 [5]. Due to large-scale antimalarial interventions, malaria incidence decreased gradually, and malaria elimination was achieved in 2019 in Hainan, with no local cases reported for six consecutive years from 2016 to 2021.

Currently, significant attention is being paid to preventing malaria reestablishment post-elimination in Hainan. However, recent data on the distribution, bionomics, and control of malaria vectors in Hainan are rare. This paper attempts to summarize the surveillance and control of malaria vectors from 1950 to 2021, from the hyperendemic to post-elimination periods, using a retrospective method, with the goal of providing scientific evidence to prevent malaria reestablishment in Hainan. The purpose of this paper is to (1) review malaria vector-related surveillance and control in Hainan; (2) identify knowledge gaps for research in the future; and (3) discuss vector control strategies that may help to prevent malaria reestablishment due to imported malaria in Hainan.

## 2. Materials and Methods

PubMed and CNKI (China National Knowledge Infrastructure) were used to carry out an electronic search of the literature published in English and Chinese languages between January 1950 and December 2021 (Figure 1). The terms ‘*Anopheles*, Hainan’ were used to search. Secondary data from one published book [6] and two unpublished books [7,8] were used for our review. Malaria vector-related epidemiological terms that are important to malaria control and elimination in Hainan were chosen for the search, including species and distribution, infection by *Plasmodium* parasite, vectorial capacity, seasonality, bionomics, resistance to insecticide, and vector control. Literature screening will begin with the removal of duplicated studies. Studies that do not contain malaria vector-related terms in Hainan in the titles and abstracts will be excluded during title and abstract screening. The relevant full-text articles were further reviewed and studies explicitly containing malaria vector-related epidemiological terms in Hainan (including species and distribution, infection by *Plasmodium* parasite, vectorial capacity, seasonality, bionomics, resistance to insecticide, and vector control) were selected in the final list of review (Figure 1).

According to malaria prevalence in Hainan Province from 1950 to the present, the course of malaria control in Hainan can be divided into five phases: the severe epidemic phase (1950–1980); the slowly declining phase (1981–2000); the steadily declining phase (2001–2010); the elimination phase (2011–2020); and the post-elimination phase (2020 to present) (Table 1).

Severe epidemic phase (1950–1980): In this phase, malaria was hyperendemic, with annual malaria incidence higher than 100 cases per 10,000 people in most years. Most malaria infections occurred in villages and *P. falciparum* was the main parasite in this phase. As data on prevalence, parasites, and vectors were limited in this early phase, significant research on species, distribution, seasonality, blood preference, and vectorial capacity was conducted to guide control strategies. Large-scale IRS with DDT was used to control primary vectors in high-risk areas during the transmission seasons of each year (April to May, and August to September).

Slowly declining phase (1981–2000): In this phase, malaria cases were slowly declining. Most malaria infections occurred in mountainous areas in the southern and central areas of the province. *P. vivax* was the main parasite species at this phase. Large-scale IRS with DDT or pyrethroid insecticides and ITNs with pyrethroid insecticides were used to control primary vectors in high-risk areas during the transmission seasons of each year.

Steadily declining phase (2001–2010): In this phase, malaria cases were steadily declining and only six deaths were reported, although a rebound was recorded from 2002 to 2004. IRS in foci, ITNs, or LLINs with pyrethroid insecticides were used to protect high-risk populations, especially populations that stay overnight in mountainous areas.

Elimination phase (2011–2020): In this phase, reported malaria cases fell further. Most of the reported malaria cases were imported cases. IRS in foci and ITNs or LLINs with pyrethroid insecticides were used to protect high-risk populations and interrupt malaria transmission in local areas.

Post-elimination phase (2020 to the present): In this phase, vector surveillance is still necessary to guide vector intervention strategies. IRS with pyrethroid insecticides is used to control malaria in foci with primary vectors in this phase.

## 3. Results

Our initial search yielded 273 records, of which 248 remained after the removal of duplicates (Figure 1). In total, 79 papers and reports met our criteria. The other literature was excluded after screening the title and abstract or reviewing the full text, as they did not contain data or information on malaria vector-related ecology, bionomics, or control. Table 2 categorizes the 79 papers. Six references dealt with infection by *Plasmodium* parasites, six with vectorial capacity, 41 with species and distribution, seven with seasonality, eight with bionomics, 13 with resistance to insecticides, and 14 with vector control. In the last 10 years (2012–2021), 16 papers have been published on malaria vectors, of which two papers address seasonality and vector control in Hainan.

Figure 2 showed that more studies were published from 1991 to 1999, with 50% (17/34) of published papers addressing species and distribution. However, the number of published papers decreased, and only 19 papers were published from 2012 to the present, 57.9% (11/19) and 31.6% (6/19) of the published papers were studies on species and distribution, and resistance to insecticides, respectively.

**Table 2 tropicalmed-08-00131-t002:** Summary of selected references.

Category	Sub-Category	References				
		1950–1980	1981–1990	1991–1999	2000–2011	2012–Present
Species and distribution (41)		[9]	[3]	[10,11,12,13,14,15,16,17,18,19,20,21,22,23,24,25,26]	[5,27,28,29,30,31,32,33,34,35,36]	[37,38,39,40,41,42,43,44,45,46,47]
Infection by *Plasmodium* parasite (6)		[48]	[49]	[16,50,51]	[4]	-
Vectorial capacity (6)		-	[52,53,54]	[23,55]	[4]	-
Seasonality (7)		[48]	[52]	[16,55]	[4,56]	[44]
Bionomics (8)	Blood preference (3)	-	[52,57]	[16]		-
	Nocturnal activity (4)	-	-	[16,58]	[4,59]	-
	Flight distance (2)	-	[60,61]	-	-	-
Resistance to insecticides (13)		[62]	[52]	[16,63]	[64,65,66]	[45,67,68,69,70,71]
Vector control (14)		[48]	[52,72]	[16,73,74,75,76,77]	[4,78,79,80]	[37]

Numbers in this table are the total number of references for each category. Numbers in brackets correspond to the reference citation.

### 3.1. Species and Distribution

A total of 32 species of anopheline mosquitoes have been recorded in Hainan Province [10]. *An. dirus* and *An. minimus* have been confirmed as the principal malaria vectors in Hainan Province, according to a biological investigation of various anopheline mosquitoes and the epidemiological characteristics of malaria. These two species meet all criteria of vector incrimination in Hainan, including (1) evidence that temporal and spatial distribution of local human malaria cases is dependent on *An. dirus*/*An. minimus* [4,7,59], (2) evidence of a high degree of anthropophily in *An. dirus*/*An. minimus* [7,44,48,57], (3) evidence that *Plasmodium* sporozoites can be harbored in the salivary glands of *An. dirus*/*An. minimus* under natural conditions [4,7,16,48], and (4) evidence that the *Plasmodium* parasite harbored by *An. dirus*/*An. minimus* can be transmitted under experimental conditions [49,50,51].

*An. sinensis*, *An. candidiensisand* and *An. philippinensis* harbor malaria sporozoites in their salivary glands in Hainan, with sporozoite rates of 0.02, 0.21, and 0.06, respectively [7]. However, they are considered secondary malaria vectors because they have a lower degree of anthropophily in Hainan.

*An. lesteri*, one of the main malaria vectors in other provinces in China, is distributed in Wenchang, Haikou, and Wanning. It is not a malaria vector in Hainan, as no malaria sporozoites were found in its salivary glands [8,14,17,42,43,46,47].

#### 3.1.1. *An. dirus*

*Anopheles dirus s.s.* and *Anopheles baimaii*, belonging to the *Anopheles dirus* complex, are important malaria vectors [81]. Results from previous research indicate that *An. dirus* in Hainan is a specimen of the species *Anopheles dirus s.s*. [18,19,26,30].

*An. dirus* is an exophilic mosquito and prefers to suck human blood [82]. It accounts for 0.02% to 0.54% of the composition of *Anopheles* mosquitoes in human rooms in the daytime [82]. However, it accounts for 3.8% of the composition of *Anopheles* mosquitoes when human plus cattle bait was used in the nighttime [7].

In Hainan Province, its larvae breed in riverbeds and rocky caves where there is shade, and decaying leaves [7]. A survey on the distribution of *An. dirus* from 1965 to 1975 indicated that *An. dirus* was mainly distributed in mountainous areas, especially in the south-central part of Hainan Island [11,12,25,26,31,33,34,35,40]. Due to its breeding habits, environmental modifications, such as cutting down shrubbery, or land use change can lead to the reduction in *An. dirus* [15,18,22,25,28,38,40,41].

Results on malaria vectors over the last 70 years indicate that the population and distribution of *An. dirus* are decreasing [5,15,20,25,28,29,38,40,41]. Longitudinal surveillance in Qiongzhong shows a significant decreasing trend in the population of *An. dirus*, with 1.67 per night in 2005 to 0.03 per night in 2014 [44]. It is now difficult to catch *An. dirus* at this site. In another surveillance site in Wuzhishan, the population of *An. dirus* decreased from 1.17 per night in 2005 to 0.33 per night in 2014 [44]. Vector surveys during seasonal anti-malarial programs showed a gradual and significant decrease in the presence of *An. dirus* in Hainan (Figure 3). It means that the spatial distribution of *An. dirus* is shrinking. The number of cities/counties with *An. dirus* decreased from 10 in 1981–1990 tofour during 2011–2020 (Figure 4). The decrease in the *An. dirus* population may lead to the disappearance of this species in certain counties and cities in Hainan. At present, *An. dirus* is mainly distributed in mountainous areas, including Changjiang, Wuzhishan and Qiongzhong.

#### 3.1.2. *An. minimus*

*An. minimus* can be classified into A and B types in Hainan, based on ecological and morphological surveys and controlled trials [7,57]. Type A is an endophilic mosquito that prefers to rest in living rooms and is extremely sensitive to DDT [16]. However, type B is an exophilic mosquito that prefers to rest outside and avoids surfaces with DDT, but demonstrates excitement when contacting DDT. Results from PCR testing indicated that *An. minimus* populations from Wenchang and Wanning were *An. minimus* A [27].

*An. minimus* prefers human and cattle blood. *An. miminus* is the dominant species of domestic mosquito, accounting for 37.1% of anopheline mosquitoes in human rooms in the daytime [7].

*An. minimus* breeds in clear water bodies, with less humus shaded by plants, such as small streams, slow-flowing rivers, irrigation ditches with water plants, stagnant water in riverbeds, and so on [16]. *An. minimus* was previously widely distributed in mountainous, hilly, and coastal plain areas over the whole province in the 1950s [7,9].

*An. minimus* was previously distributed across Hainan Island, but the presence of *An. minimus* has changed dynamically [10,11,13,15,20,21,23,24,27,28,31,32,33,34,35,36,37,38,39,41,44,45]. Since the use of IRS with DDT to control malaria mosquitoes in 1959, the distribution of *An. mimimus* has shrunk dramatically. A survey from 1963 to 1965 showed that *An. minimus* had been almost eradicated [3]. However, results from a survey in 1978 showed that *An. minimus* had rebounded (Figure 5) [3,7]. However, the presence of *An. minimus* was at a stable rate of 11% to 26% in most years after 1978 (Figure 5). It means that *An. minimus* is hard to eliminate naturally in a given region although there was a greater improvement in socio-economic development and large-scale implementation of vector control interventions. Vector surveys during the seasonal anti-malarial program from 1980 to 2020 indicate that the percentage of presence of *An. minimus* at various sites is around 30%, and the number of counties/cities with *An. minimus* decreased from 12 in 1980–1989 to nine in 2010–2020, mainly distributed in the south-central and western areas of Hainan Province (Figure 4). To further clarify the distribution of *An. minimus* in Hainan Province, a special survey was conducted throughout the island from 2011 to 2015. Results showed that among 56 survey sites in 18 cities and counties, *An. minimus* was distributed in 17 survey sites in nine cities/counties (Figure 6). *An. minimus* was not recorded in coastal plain areas but in hilly and mountainous areas during this survey.

Studies on the species and distribution of primary malaria vectors are the main research topic in Hainan. In certain areas, data on species and the distribution of primary malaria vectors can provide scientific evidence for the implementation of vector control interventions. Vector control interventions should be implemented to stop malaria transmission in malaria foci with primary vectors. In Hainan, vector control interventions should be implemented to stop malaria transmission when malaria cases are reported in the central and southern mountainous areas of Hainan, where the primary malaria vectors *An. minimus* and *An. dirus* are still present.

### 3.2. Infection by Plasmodium Parasites

In laboratory tests, *An. dirus* from the Hainan population demonstrated high infection rates of *P. vivax*, and *Plasmodium cynomoig* [49,50,51]. From 1951 to 1965, field surveys of natural infection of the salivary glands of 67,676 anopheline mosquitoes (including *An. minimus*, *An. dirus*, *An. fluviatilis*, *An. sinensis*, *An. jeyporiensis*, *An. philippinensis*, *An. arbirostris*, *An. kochi*, *An. aconitus*, *An. tessellates*, *An, peditaeniatus*, *An. Splendidus,* and *An. annularis*) were conducted in Hainan Province. Six species of anopheline were infected by *Plasmodium* parasites. The proportions of *An. minimus*, *An. dirus*, *An. fluviatilis*, *An. sinensis*, *An. Jeyporiensis,* and *An. philippinensis* infected with *Plasmodium* parasites were 4.31% (1975/45815), 2.46% (77/3133), 2.35% (17/724), 0.02% (1/5050), 0.21% (15/7317) and 0.06% (1/1740), respectively [7,16,48]. However, research on *Anopheles* mosquitoes infected by *Plasmodium* parasites in the field is limited, aside from a report showing that the infection rate in *An. dirus* was 4.6% in Qiongzhong in 1990–1994 [4].

Studies on *Plasmodium* parasite infection in malaria vectors in Hainan are limited, with the most recent studies carried out in the 1990s [4,16,49,50,51]. Knowledge of infection of malaria vectors by *Plasmodium* parasites is essential to determine the primary vectors. In Hainan, *An. minimus*, *An. Dirus,* and *An. fluviatilis* have demonstrated a higher *Plasmodium* infection rate, while *An. sinensis*, *An. jeyporiensis*, *and An. philippinensis* have demonstrated a lower rate.

### 3.3. Vectorial Capacity

Vectorial capacity (VC) is important for assessing local malaria prevalence, estimating the effectiveness of vector control interventions, and forecasting the risk of reestablishment post-elimination. In Hainan, previous research has provided estimates of the VC of *An. dirus* and *An. minimus* (Table 3) [4,23,52,53,54,55]. In general, *An. dirus* is often considered to be a more efficient vector compared to *An. minimus* in Hainan. A longitudinal survey on the average VC of *An. dirus* in Wanning from 1983 to 1985 found values of 0.9, 0.4, and 0.2, while the highest monthly vectorial capacities of *An. dirus* were 4.5, 2.7, and 1.8, respectively [53]. Another survey in Baoting from March 1992 to June 1992 indicated that the highest monthly vectorial capacity of *An. dirus* was 6.2 [55]. The highest monthly vectorial capacity of *An. minimus* was 0.34 in Baisha between April 1983 and June 1983 [52].

Studies on the vectorial capacity of primary malaria vectors in Hainan are dated, with the most recent studies having been carried out during the 1980s to 1990s [4,23,52,53,54,55]. The greater anthropophily and increased survival of *An. dirus* may contribute to its higher vectorial capacity (Table 3).

### 3.4. Seasonality

#### 3.4.1. *An. dirus*

Due to the characteristic breeding sites of *An. dirus*, its seasonality is closely associated with the amount of rainfall [4,44,55,56]. Figure 7 shows the seasonality of *An. dirus* in Hainan from 2001 to 2020. A single peak is recorded in one year, with the peak varying with the time of year and region. For example, the seasonal peak of *An. dirus* in Wuzhishan was from September to November in 1964, while it was from June to July in 1965 [7]. In 1984, the seasonal peaks of *An. dirus* were from May to June, November, and July to August in Wuzhishan, Wanning, and Baisha, respectively [8]. Data from 2001–2020 show that the seasonal peaks of *An. dirus* was from April to May in Wuzhishan, and from May to September in Qiongzhong (Figure 7).

#### 3.4.2. *An. minimus*


Figure 8 shows the seasonality of *An. minimus* in Hainan from 2002 to 2016. The seasonal peak of *An. minimus* is from April to June in Hainan but varies slightly between areas [16,48]. For example, a survey in Danzhou in the 1950s showed that the population of *An. minimus* started to increase in March, peaked from April to June, and gradually declined after July [16]. Another smaller peak was observed from September to October in Changjiang and Wanning [7,56]. Longitudinal seasonal data in Changjiang, Baoting, Wenchang, Sanya, and Dongfang from 2001–2014 show that the seasonal peaks of *An. minimus* were from April to June (Figure 8). The population increases at the beginning of the rainy season. This may be related to the fact that rainfall provides a large number of breeding sites at the beginning of the rainy season. However, the population decreases as breeding sites are washed away due to excessive rainfall after July.

Data on the seasonality of *An. dirus* and *An. minimus* can provide scientific evidence for the timing of large-scale vector control implementation. Interventions should be taken before the seasonal peak to suppress the abundance of primary vectors and lower the risk of being bitten by infected mosquitoes.

### 3.5. Bionomics

#### 3.5.1. Blood Preference of *An. minimus*

The blood preference of *An. minimus* changed from human blood to both human and cattle blood. A study identifying the origin of mosquito blood meals using enzyme-linked immunosorbent assay was conducted in Hainan Province from 1955 to 1964. The results indicate that *An. minimus* prefers human blood when resting in a house [7]. Similar results were reported in Baisha in 1978 [52]. However, due to the continuous implementation of IRS with DDT to control *An. minimus*, the blood meal preference of *An. minimus* changed. A study in 1984 indicated that *An. minimus* prefer cattle blood in Changjiang [16,57]. At present, *An. minimus* in hilly areas has similar preferences for cattle and human blood (e.g., Danzhou, Baisha). However, populations in mountainous areas prefer cattle blood (e.g., Dongfang).

Blood feeding preference alters host-vector-pathogen contact rate and hence the transmission dynamic of mosquito-borne diseases via its impact on R_0_ depending on whether a preferred/avoided host can be a competent pathogen reservoir and/or a biting mosquito can be a competent vector [83,84]. Blood feeding preference can be affected by myriad determinations, such as odorants, body mass of the host, physiology and genetics of the mosquito, host abundance, etc. [83,84]. However, studies on the factors that contribute to changes in the blood preference of *An. minimus* are limited in Hainan.

#### 3.5.2. Nocturnal Activity

The nocturnal activity of *An. dirus* in Hainan peaks from 10 p.m. to 2 a.m. the next day, and stop after 5 a.m. [7]. *An. dirus* can bite a human in dark forests during the daytime [4]. In addition, the nocturnal activity of *An. dirus* may be associated with the phase of the moon [7,58,59]. A survey in Dongfang showed that more *An. dirus* were captured before midnight than after midnight when the moon was in its first quarter, while the opposite was true when the moon was in its last quarter [58]. However, other studies in Wuzhishan, Wanning, and Baisha in 1984–1985 indicated that the activity of *An. dirus* was not associated with the phase of the moon [8]. Similar numbers of *An. dirus* were caught during new moons and full moons.

Research on the nocturnal activity of *An. minimus* is limited, aside from a review in 1983 showing that *An. minimus* starts to bite at 10 p.m. and reached a biting peak at 11 p.m. in Hainan [16].

Nocturnal activity is very important in determining times when personal protection is needed. Studies on nocturnal activity indicate that at-risk populations (e.g., rubber-tapping workers or people who stay overnight in mountains) should spray DEET when working at night and use bednets when sleeping outside to reduce the risk of being bitten by infected mosquitoes.

#### 3.5.3. Flight Distance of *An. dirus*

In 1983, the mark-release-recapture method was used to observe the flight distance of *An. dirus* in Qiongzhong. A total of 30,052 *An. dirus* (half male and half female) were marked and released, while 71 *An. dirus* (all female) were recaptured. The flight directions covered all four points of the compass and were at least 1 km, with a maximum flight distance of 1.4 km to the south [61]. In 1985, 22,935 *An. dirus* were released again and 404 were recaptured, of which one was a male and the rest were female. Of the recaptured *An. dirus*, 97% were within 1 km of the release site. The maximum flight distance was 1.6 km to the south [60].

Studies on the flight distance of *An. dirus* are essential for malaria foci disposal, especially when local malaria cases are reported in certain malaria-free areas. Populations living within 1 km of foci are at high risk.

### 3.6. Resistance of Anopheline Mosquitoes to Insecticides

Although few studies have reported on the resistance of *An. minimus* and *An. dirus* to insecticides, existing studies indicate that the two primary malaria vectors are susceptible to insecticides. However, studies on the resistance of *An. sinensis* to insecticides have indicated that *An. sinensis* is resistant to many insecticides, especially pyrethroids.

#### 3.6.1. *An. dirus*

The resistance of *An. dirus* to insecticides in Hainan province is summarized in Table 4. The results indicate that *An. dirus* females are susceptible to insecticides. Populations of *An. dirus* females from Wuzhishan, Wanning, and Changjiang demonstrated susceptibility to 4% DDT, 0.05% deltamethrin, and 5% malathion with 100% mortality [63,66]. Results from the larval dipping method in 2007 indicated that larvae of *An. dirus* were susceptible to deltamethrin and cyfluthrin [65].

#### 3.6.2. *An. minimus*

The resistance of *An. minimus* to insecticides in Hainan province is summarized in Table 5. The results indicate that *An. minimus* females are susceptible to insecticides. The LC_50_ of DDT to *An. minimus* females from Danzhou in 1978 and Dongfang in 1981 were 0.00055 mg/L and 0.00048 mg/L, respectively [64]. *An. minimus* females from Baisha demonstrated susceptibility to 4% DDT in 1978 [16,52], and populations from Changjiang demonstrated susceptibility to 4% DDT, 0.05% deltamethrin, 5% malathion, and 0.15% cyfluthrin in 2010 [66].

#### 3.6.3. *An. sinensis*

The resistance of *An. sinensis* to insecticides in Hainan province is summarized in Table 6. From 1960 to 1999, populations from Haikou and Dongfang demonstrated resistance to DDT, with a LC_50_ higher than 2% [62]. Populations from Danzhou demonstrated possible resistance to DDT, with a LC_50_ between 1% and 2% [64]. Populations from Baisha and Wanning demonstrated susceptibility to gamma hexachlorocyclohexane and DDT, with a LC_50_ lower than 1% [7].

The resistance of wild-caught female mosquitoes of *An. sinensis* from nine cities/counties (including Haikou, Sanya, Lingshui, Qiongzhong, Baoting, Dongfang, Baisha, Ledong, and Changjiang) to 4% DDT, 0.05% deltamethrin and 5% malathion was assayed using the World Health Organization standard resistance tube assay procedure from 2010 to 2014 [67,70,71]. Results indicated that *An. sinensis* females were resistant to 4% DDT. Possible resistance to 0.05% deltamethrin was observed in populations from Baisha and Qiongzhong. Populations from other districts showed resistance to 0.05% deltamethrin. Populations from Baisha, Qiongzhong, and Dongfang were susceptible to 5% malathion, but populations from other districts were not [65,68,70]. From 2018 to 2020, *An. sinensis* population from Sanya demonstrated susceptibility to 5% chlorfenapyr, susceptibility or possible resistance to 5% malathion, and possible resistance or resistance to 0.5% ethofenprox, 2.0% fipronil, 0.05% deltamethrin and 5% DDT (unpublished data).

Studies on the mechanism of *An. sinensis* to insecticides are limited in Hainan, with only two reports from 2014 [68,69]. Results from these studies indicate that the mutation frequency of kdrL1014F is low (less than 10%), but the mutation frequency of ACE-1G119S is high (45–75%). In addition, P450 monooxygenase was highly expressed in *An. sinensis* populations that demonstrated resistance to deltamethrin. The results indicate that the resistance of *An. sinensis* in Hainan is mainly related to the high expression of detoxifying enzymes [69].

*An. dirus* and *An. minimus* remain susceptible to commonly used insecticides, which may be related to lower exposure to insecticides in the breeding sites of these two major vectors. However, extensive resistance of *An. sinensis* to insecticides has been observed. Insecticides have not been used directly to control *An. sinensis*, but larvae of *An. sinensis* are exposed to more insecticides because one of their main breeding sites is paddies, where insecticides are used to control other bugs.

The most recent studies on the resistance of *An. dirus* and *An. minimus* to insecticides were carried out before 2010. The current resistance level of the two primary vectors to insecticides is concerning, however, it is not being reported because it is hard to breed enough F1 progeny, as the field populations of *An. dirus* and *An. minimus* are low at present [44]. Further investigations on the identification of resistance mechanisms are limited, with two published papers [68,69].

**Table 6 tropicalmed-08-00131-t006:** Resistance of *Anopheles sinensis* to insecticides in Hainan.

Method	Year	Insecticide	Population	LC_50_ (mg a.i./L) or LT_50_ (min)or KT_50_ (min) (95% Confidence Interval) or Knockdown Rate after 1 h Exposure (%)	Toxicity Regression Line/ Mortality after 24 h Exposure	Resistance Index/ Resistance Level	Reference
WHO tube method	1960–1999	DDT	Haikou	LC_50_ > 20 mg/L	-	Resistance	[64]
	1960–1999		Danzhou	LC_50_ = 10–20 mg/L	-	Possible resistance	[64]
	1960–1999		Baisha	LC_50_ < 10 mg/L	-	Sensitive	[64]
	Unknown		Danzhou	LC_50_ = 16.2 mg/L	-	Possible resistance	[64]
	Unknown		Baisha	LC_50_ = 9.2 mg/L	-	Sensitive	[64]
	Unknown	Hexachlorocyclohexane	Baisha	LC_50_ = 0.4 mg/L	-	Sensitive	[64]
WHO tube method at diagnosis dose	1961	4% DDT	Wanning	-	Mortality = 88%	Sensitive	[64]
	2010	4% DDT	Dongfang	Knockdown rate = 2%	Mortality = 19.8%	Resistance	[66]
		0.05% deltamethrin		Knockdown rate = 2%	Mortality = 22.9%	Resistance	[66]
		5% malathion		-	Mortality = 43.8%	Resistance	[66]
	2011	0.05% deltamethrin	Haikou	-	Mortality = 35%	Resistance	[70]
		4% DDT		-	Mortality = 36%	Resistance	[70]
		5% malathion		-	Mortality = 39%	Resistance	[70]
	2012	0.05%Deltamethrin	Sanya	-	Mortality = 25.7%	Resistance	[70]
		4% DDT		-	Mortality = 27%	Resistance	[70]
		5% malathion		--	Mortality = 16%	Resistance	[70]
		0.05% deltamethrin	Lingshui	-	Mortality = 17%	Resistance	[70]
		4% DDT		-	Mortality = 24%	Resistance	[70]
		5% malathion		-	Mortality = 41%	Resistance	[70]
	2013	0.05% deltamethrin	Qiongzhong	-	Mortality = 95%	Possible resistance	[70]
		4% DDT		-	Mortality = 60.9%	Resistance	[70]
		5% malathion		-	Mortality = 100%	Sensitive	[70]
		0.05% deltamethrin	Ledong	-	Mortality = 38%	Resistance	[70]
		4% DDT		-	Mortality = 41%	Resistance	[70]
		5% malathion		-	Mortality = 56%	Resistance	[70]
	2014	0.05% deltamethrin	Baoting	-	Mortality = 68.9%	Resistance	[70]
		4% DDT		-	Mortality = 53%	Resistance	[70]
		5% malathion		-	Mortality = 73%	Resistance	[70]
		0.05% deltamethrin	Dongfang	-	Mortality = 49%	Resistance	[70]
		4% DDT		-	Mortality = 31%	Resistance	[70]
		5% malathion		-	Mortality = 99%	Sensitive	[70]
		0.05% deltamethrin	Baisha	-	Mortality = 94%	Possible resistance	[70]
		4% DDT		-	Mortality = 72%	Resistance	[70]
		5% malathion		-	Mortality = 99%	Sensitive	[70]
	2018	0.5% etofenprox	Sanya	Knockdown rate = 90%	Mortality = 80%	Resistance	Unpublished data
		5% malathion		Knockdown rate = 31%	Mortality = 91%	Possible resistance	Unpublished data
		2.0% fipronil		Knockdown rate = 30%	Mortality = 77%	Resistance	Unpublished data
		0.05% deltamethrin		Knockdown rate = 14%	Mortality = 43%	Resistance	Unpublished data
		4% DDT		Knockdown rate = 30%	Mortality = 71%	Resistance	Unpublished data
		5% chlorfenapyr		Knockdown rate = 69%	Mortality = 100%	Sensitive	Unpublished data

LC_50_, Lethal concentration 50% (the concentration expected to kill exactly 50% of exposed mosquitoes). LT_50_, Lethal time 50% (the time expected to kill exactly 50% of exposed mosquitoes at a certain dose). KT_50_, Knockdown time 50% (the time expected to knockdown exactly 50% of exposed mosquitoes at a certain dose).

### 3.7. Vector Control

According to the meteorological conditions and characteristics of malaria transmission in Hainan, IRS and ITNs are used to control malaria vectors during seasonal anti-malarial programs (in April and August every year) (LLINs have been distributed for malaria control since 2007).

Since the 1950s and 1960s, Hainan has carried out IRS in malaria hyperendemic and endemic areas [48,52,73,78,80]. In 1984, deltamethrin-treated nets were applied in a small area of the province to kill mosquitoes [8]. Pyrethroid insecticides have gradually become the main chemical for mosquito control in seasonal anti-malaria programs since 1991 [4,6,8,16,37,74,75,76,79]. Figure 9 shows the number of residents protected by malaria vector intervention in seasonal anti-malarial programs from 1986 to 2020. An estimated 6.39 million people were protected by insecticides from 1986 to 2020 (Figure 9). The number of residents protected by malaria vector intervention decreased from 0.33 million in 1993 to 0.09 million in 2006. However, it increased to its peak in 2009 (0.52 million) and declined to 0.003 million in 2020.

To our knowledge and according to the results of our review, IRS and ITNs (including LLINs) are used for malaria vector control in Hainan. Four previous papers have studied the effect of IRS or ITNs with deltamethrin or permethrin or IRS with DDT on the population of malaria vectors and the incidence of malaria [52,75,76,77]. Three of these papers indicated that IRS with DDT and ITNs with deltamethrin and permethrin can reduce the population of *An. minimus* and *An. dirus*, and therefore malaria incidence, significantly [52,75,77]. However, a study by Li et al. [76] showed that the population of *An. dirus* and malaria incidence were not reduced significantly after treatment using ITNs with deltamethrin. One reason for this is that local residents moved around the village before going to bed at night [76]. This study indicates that supplementary interventions, such as health education, should be undertaken to enhance the effect of malaria vector control. LLINs have been used since 2007, however, the effect of LLINs has not been formally evaluated under natural conditions in Hainan.

Two papers published in 1989 and 2002 described the effect of land-use change on populations of malaria vectors [72,77]. With the change of shrubs to cash crops or improvement of living conditions (traditional homes with mud walls, or thatched roof houses to burnt-brick wall and tile-roofed houses), the population of *An. dirus* decreased dramatically, leading to the reduction in malaria in the local area.

Most studies on vector control in Hainan have focused on the effects of IRS and ITNs (LLIN). There are limited studies on space spraying in Hainan.

## 4. Discussion

We reviewed malaria vector research and control activities occurring across 70 years in Hainan, from 1951 to 2021. However, entomology research in Hainan is limited, with only 13 published studies and three scientific reviews available from the last 10 years [37,38,39,40,41,42,43,44,45,46,47,67,68,69,70,71]. As environmental landscapes continue to change and socio-economic development continues to improve, results and conclusions on malaria vectors from previous literature in Hainan may not always be correct. Malaria vector-related surveillance and control should be continuously conducted to provide scientific evidence for preventing malaria reestablishment by imported cases, post-elimination in Hainan.

*Anopheles minimus* and *Anopheles dirus* are the primary malaria vectors in Hainan. To effectively prevent malaria reestablishment by imported cases, it is important to understand the distribution of these vectors and their ability to sustain malaria transmission. However, the literature on malaria vector distribution in Hainan is somewhat limited and out of date (Figure 4 and Figure 6) [42,43,44]. The distribution of malaria vectors may have changed with changes in the environmental landscape, however, these data have not been reported in the last eight years.

*Anopheles dirus* has higher VC than *An. minimus* and is considered a more important vector compared to *An. minimus* in Hainan [4,7,8,22,52,53,54,55]. From 1985 to 2011, most local malaria cases were reported in populations who stay overnight in mountainous areas where *An. dirus* is present [6,8]. This information is important to classify the risk level of malaria reestablishment. Areas such as Wuzhishan and Qiongzhong where *An. dirus* is still present are considered to be at high risk of malaria reestablishment. Areas where *An. minimus* is present are considered to be at medium risk of malaria reestablishment. Areas without *An. dirus* and *An. minimus* are considered to have a low risk of malaria reestablishment.

The bionomics of anthropophily and the peak biting times of local malaria vectors should be considered carefully to effectively select malaria vector interventions. As *An. dirus* is an outdoor biter with high anthropophily [4,7,8,22,52,53,54,55], effective interventions are the use of LLINs or ITNs to protect residents who stay overnight in mountainous areas. On the other hand, in Hainan, *An. minimus* exhibits outdoor biting behavior and lower anthropophily [16]. The impact of local alternate hosts on the biting behavior of malaria vectors is complicated [85]. The blood meal preference of mosquitoes may change from humans to animals, possibly leading to less direct contact with mosquitoes, or increasing the biting population of mosquitoes and the risk of being bitten. Furthermore, biting behavior may change seasonally [86].

It is important for policy-makers to understand the resistance of malaria vectors to insecticides in order to choose effective insecticides for malaria vector control. Whether during large-scale malaria vector intervention or in vector control-related disposal at malaria foci, the resistance of malaria vectors to insecticides should be surveilled. However, data on the resistance of malaria vectors are limited, especially for *An. dirus* and *An. minimus*. The latest research on the resistance of *An. dirus* and *An. minimus* was conducted in 2009, these vectors demonstrated susceptibility to DDT, malathion, and deltamethrin. Studies in Myanmar in 2019 indicated that *An. minimus* demonstrated possible resistance to deltamethrin [87], however, *An. dirus* in Thailand was shown to be completely susceptible (100% mortality) to transfluthrin [88].

We searched the PubMed database by the term ”plant Anopheles China”, and searched China’s national knowledge infrastructure (CNKI) database by the search term “plant Anopheles” to screen studies on the effect of herbal medicine on malaria vectors in China. No published papers were selected from the PubMed database, and there were two published papers that studied the effect of herbal medicine on malaria vectors [89,90]. One study in Yunnan Province, China indicated that the not significant repellent effect of two traditional plant repellents *Artemesia argi* and *Eucalypus robusta* were observed in the field [89]. A study in Shanxi Province indicated *Tribulus terrestris*, *Biota orientalis*, and *Artenmisia annua* could kill *Anopheles* mosquito with 87–92% mortality [90].

One of the major challenges of vector control in Hainan is to select effective methods to prevent malaria reestablishment post-elimination. New methods for malaria vector control should be considered for implementation in Hainan, as some studies in other countries have indicated there are methods that can significantly reduce mosquito populations. These were included: (1) attractive toxic sugar baits, which exploit the need for both male and female mosquitoes to take sugar meals [91,92]; (2) new insecticides such as broflanilide applied for IRS [93,94,95]; and (3) treating cattle with ivermectin to reduce the mosquito population in certain areas [96].

## 5. Conclusions

This retrospective review focuses on vector-related aspects of malaria transmission and vector control in Hainan. The conditions for malaria transmission still exist in Hainan, as malaria cases might be imported from other countries and primary malaria vectors are still present. Surveillance of malaria vectors should be continually conducted to optimize vector control strategies, as the bionomics and ecology of malaria vectors can change according to environmental changes. Therefore, the role of entomology in preventing malaria reestablishment by imported cases remains crucial for Hainan.

## Figures and Tables

**Figure 1 tropicalmed-08-00131-f001:**
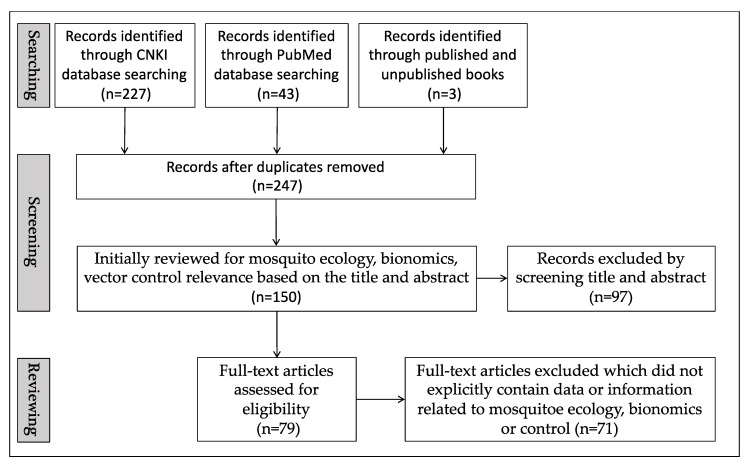
Search flowchart.

**Figure 2 tropicalmed-08-00131-f002:**
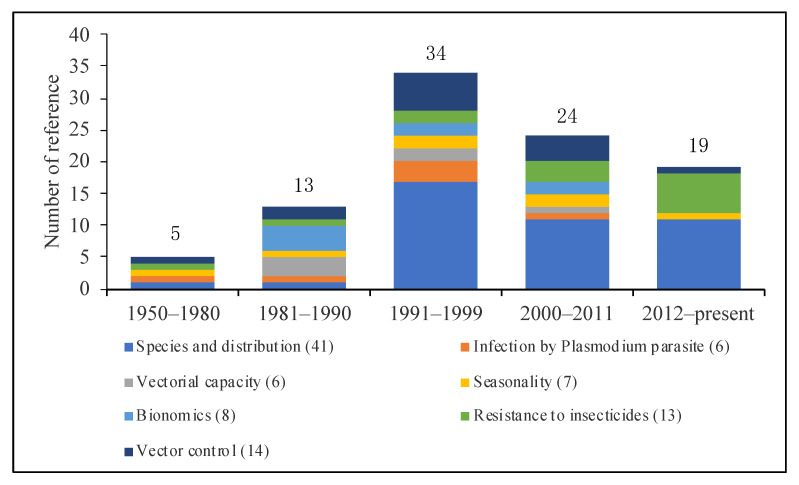
Number of references published from 1950 to the present (regardless of the repeat in different categories).

**Figure 3 tropicalmed-08-00131-f003:**
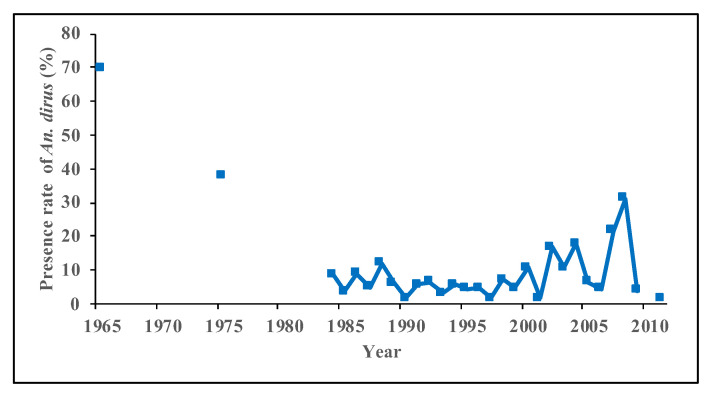
Presence of *Anopheles dirus* in Hainan Province in seasonal anti-malarial programs from 1965 to 2011.

**Figure 4 tropicalmed-08-00131-f004:**
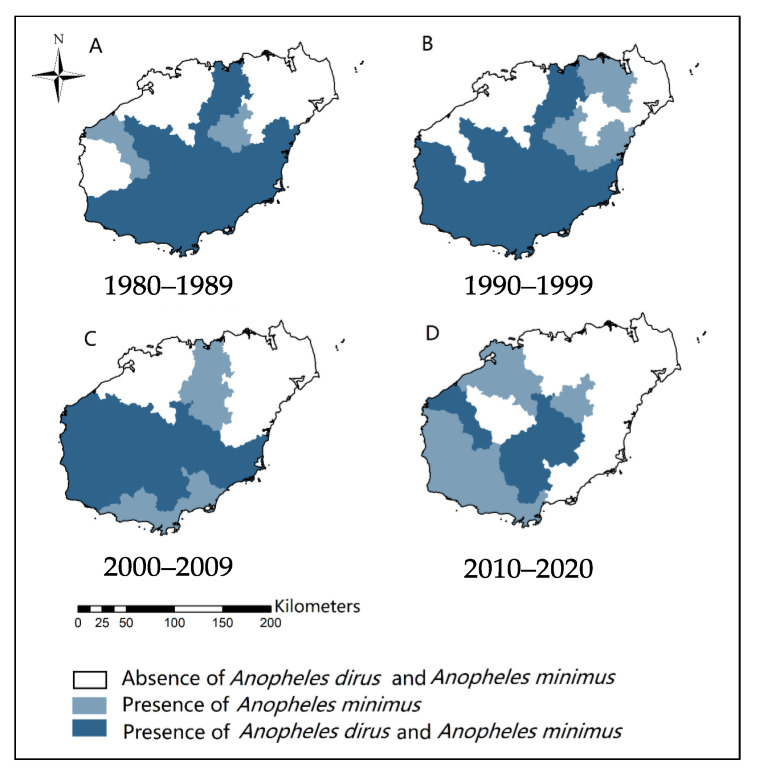
Presence of *Anopheles minimus* and *Anopheles dirus* in Hainan Province in seasonal anti-malarial programs from 1980 to 2020. (**A**) From 1980 to 1989; (**B**) from 1990 to 1999; (**C**) from 2000 to 2009; (**D**) from 2010 to 2020. The light blue color means presence of *Anopheles minimus*, and the dark blue color means the presence of both kinds of *Anopheles*.

**Figure 5 tropicalmed-08-00131-f005:**
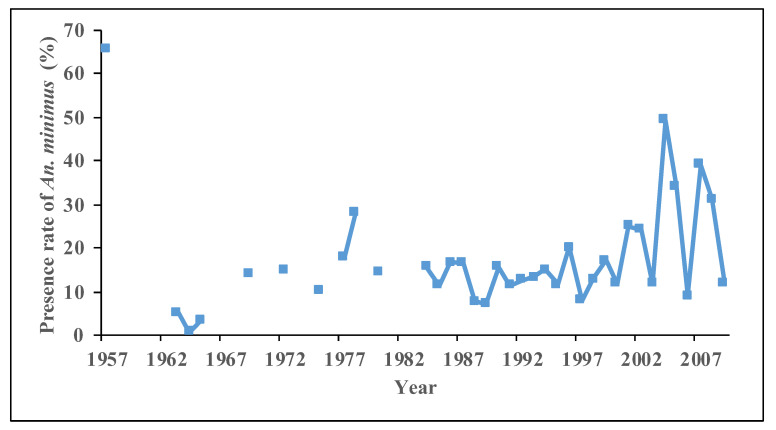
Presence of *Anopheles minimus* in Hainan Province in seasonal anti-malarial programs from 1965 to 2011.

**Figure 6 tropicalmed-08-00131-f006:**
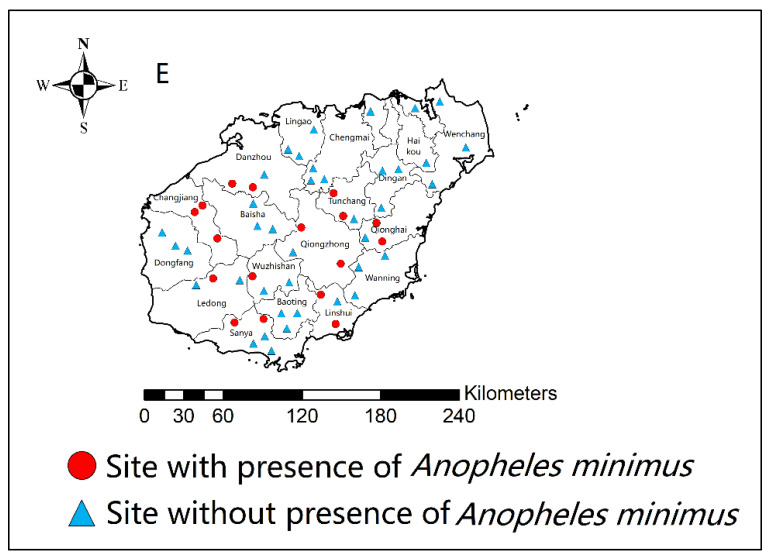
Presence and absence of *Anopheles minimus* in Hainan Province from 2011 to 2015. Red circles indicate sites with the presence of *Anopheles minimus*, while blue triangles indicate sites with the absence of *Anopheles minimus*.

**Figure 7 tropicalmed-08-00131-f007:**
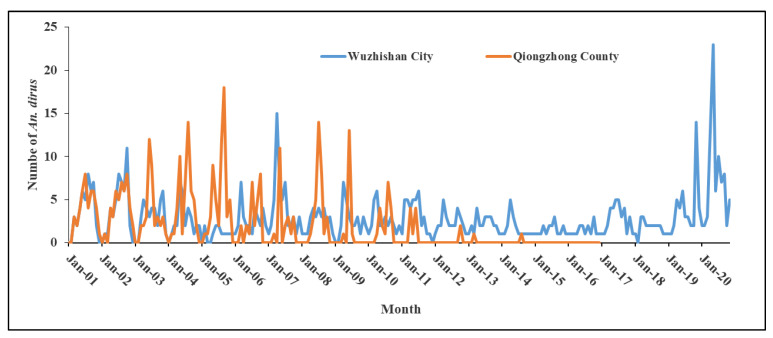
Seasonality of *Anopheles dirus* in Hainan from 2001 to 2020.

**Figure 8 tropicalmed-08-00131-f008:**
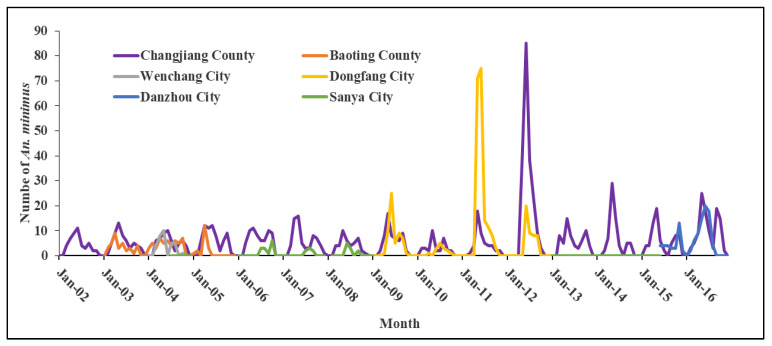
Seasonality of *Anopheles minimus* in Hainan from 2002 to 2016.

**Figure 9 tropicalmed-08-00131-f009:**
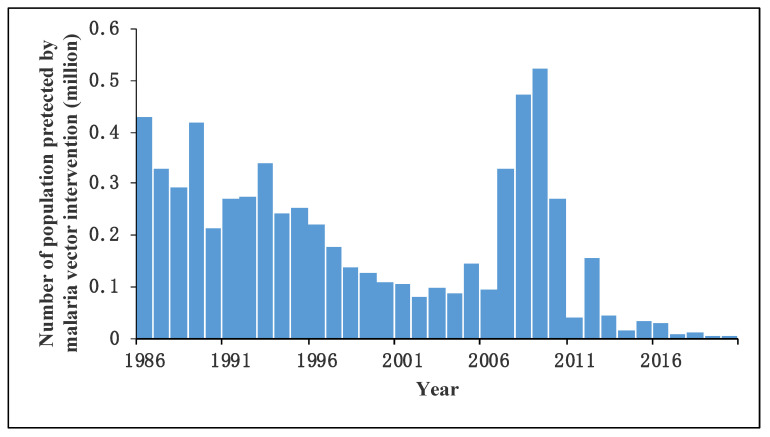
The number of residents protected by malaria vector intervention in seasonal anti-malarial programs in Hainan Province from 1986 to 2020.

**Table 1 tropicalmed-08-00131-t001:** Malaria phases and surveillance and control strategies for malaria vectors in Hainan from 1950 to the present.

Phases of Malaria Control	Approximate Number of Cases Reported	Epidemiological Characteristics	Goal	Surveillance and Control Strategies of Malaria Vectors
Severe epidemic phase (1950–1980)	1 (million)	(1) Rapidly declined with high transmission. (2) Most infections in villages. (3) Epidemic across the province. (4) *Plasmodium falciparum* was the main parasite.	To determine the prevalence of malaria, *Plasmodium* parasites, and primary malaria vectors. To reduce mortality and morbidity in hyperendemic areas.	(1) Baseline surveillance of malaria vectors. (2) Control trials. (3) Large-scale IRS with DDT was used to control primary vectors. (4) *Anopheles minimus* populations began to shrink in the 1960s but rebounded in the 1970s.
Slowly declining phase (1981–2000)	190 (thousands)	(1) Slowly declined with outbreaks. (2) Most infections in mountainous areas in the southern and central areas of the province. (3) Plasmodium vivax was the main species.	To control hyperendemic malaria.	(1) Large-scale IRS with DDT or pyrethroid insecticides and ITNs with pyrethroid insecticides to control primary vectors. (2) Vector surveillance.
Steadily declining phase (2001–2010)	40 (thousands)	(1) Steadily declined with low transmission. (2) Lower mortality (six deaths).	To maintain the reduction.	(1) IRS in foci, ITNs, or LLINs with pyrethroid insecticides for populations at high risk. (2) Vector surveillance.
Elimination phase (2011–2020)	Less than 100 (mostly imported cases)	(1) Elimination with re-emergence in mountainous areas. (2) Most malaria cases were imported.	To interrupt malaria transmission in local areas.	(1) IRS in foci and ITNs or LLINs with pyrethroid insecticides for populations at high risk. (2) Vector surveillance.
Post-elimination phase (2020 to the present)	Less than 20 cases	(1) Malaria elimination with primary vectors. (2) All the malaria cases were imported cases.	To prevent malaria reestablishment caused by imported cases.	(1) IRS with pyrethroid insecticides in foci of primary vectors. (2) Vector surveillance.

**Table 3 tropicalmed-08-00131-t003:** Vectorial capacity (VC) of *Anopheles dirus* and *Anopheles minimus* in Hainan, China.

Location	Year of Study	Species	Biting Rate (ma) ^a^	Biting Habit (a) ^b^	Survival Rate (p) ^c^	Life Expectancy of Infected Mosquito (p^n^/-lnp) ^d^	VC ^e^	Sporozoite Rate (%) (Number of Mosquitoes) ^f^	Reference
Baoting	Mar.–Jun. 1992	*An. dirus*	11.39	0.267	0.88	2.03	3.09	NC	[56]
Baisha	Apr.–Jun. 1979	*An. minimus*	1.48	0.173	0.84	1.32	0.017	Negative (416)	[57]
Wanning	1982	*An. dirus*	1.25	0.286	0.926	5.1	0.9	NC	[7]
Wanning	1983	*An. dirus*	0.95	0.286	0.926	6.6	0.9	NC	[7,58]
Wanning	1984	*An. dirus*	0.46	0.286	0.926	6.7	0.4	NC	[8,58]
Wanning	1985	*An. dirus*	0.47	0.286	0.926	3.2	0.2	NC	[8,58]
Dongfang	Jul.–Sep. 1989	*An. dirus*	4.1–11.1	0.388	0.867	1.68	1.34–3.62	2.48 (524)	[59]
Qiongzhong	1990	*An. dirus*	2.92	0.294	0.903	2.4	1.9	NC	[4]
Qiongzhong	1991	*An. dirus*	4.83	0.294	0.900	2.5	3.2	NC	[4]
Qiongzhong	1992	*An. dirus*	4.81	0.294	0.919	3.4	4.4	NC	[4]
Qiongzhong	1993	*An. dirus*	1.88	0.294	0.932	5.3	2.7	NC	[4]
Qiongzhong	1994	*An. dirus*	0.97	0.294	0.961	15.0	3.9	NC	[4]
Danzhou	1989–1990	*An. minimus*	NC	NC	NC	NC	0.029–0.683	NC	[42]
Wuzhishan	1984	*An. dirus*	0.70	0.286	NC	2.1	0.2	NC	[8]
Wuzhishan	1985	*An. dirus*	0.83	0.286	NC	14.6	1.7	NC	[8]
Baisha	1984	*An. dirus*	0.60	0.286	NC	2.5	0.2	NC	[8]
Baisha	1985	*An. dirus*	2.40	0.286	NC	2.0	0.7	NC	[8]

NC: not calculated. ^a^ Bites/person/night. ^b^ Proportion of blood meals taken from humans to total number of blood meals taken from any animal. ^c^ Probability of daily mosquito survival. ^d^ Life expectancy of infected mosquitoes. ^e^ Daily inoculations per single malaria case. VC = ma^2^ × b × p^n^/−lnp; b is the sporozoite infection rate of *Plasmodium vivax* to malaria vector under laboratory conditions. The values for *Anopheles dirus* and *Anopheles minimus* were 0.5 and 0.05, respectively. ^f^ Number of mosquitoes with sporozoites per mosquitoes tested multiplied by 100.

**Table 4 tropicalmed-08-00131-t004:** Resistance of *Anopheles dirus* to insecticides in Hainan.

Method	Year	Insecticide	Location	LC_50_ (mg a.i./L) or LT_50_ (min)or KT_50_ (min) (95% Confidence Interval) or Knockdown Rate after 1 h Exposure (%)	Toxicity Regression Line/ Mortality after 24 h Exposure	Resistance Index/ Resistance Level	Reference
Larvae dipping method	2005	Deltamethrin	Qiongzhong	LC_50_ = 0.009 (0.003~0.013) mg/L	Y = 7.617 + 1.268X	Resistance index = 1.12	[65]
			Sensitive strain	LC_50_ = 0.0068 (0.006~0.010) mg/L	Y = 9.294 + 2.015X	-	[65]
		Cyfluthrin	Qiongzhong	LC_50_ = 0.046 (0.034~0.074) mg/L	Y = 6.841 + 1.380X	Resistance Index = 1.31	[65]
			Sensitive strain	LC_50_ = 0.035 (0.028~0.044) mg/L	Y = 8.0977 + 2.126X	-	[65]
WHO tube method at diagnosis dose	1996	4% DDT	Wuzhishan	-	Mortality = 100%	Sensitive	[16]
			Wanning	-	Mortality = 100%	Sensitive	[16]
	2007	0.15% cyfluthrin	Changjiang	LT_50_ = 23.101 (19.101–27.949) min	-	Possible resistance	[66]
	2008	4% DDT		Knockdown rate = 82%	Mortality = 100%	Sensitive	[66]
		0.05% deltamethrin		Knockdown rate = 100%	Mortality = 100%	Sensitive	[66]
		5% malathion		-	Mortality = 100%	Sensitive	[66]

LC_50_, Lethal concentration 50% (the concentration expected to kill exactly 50% of exposed mosquitoes). LT_50_, Lethal time 50% (the time expected to kill exactly 50% of exposed mosquitoes at a certain dose). KT_50_, Knockdown time 50% (the time expected to knockdown exactly 50% of exposed mosquitoes at a certain dose).

**Table 5 tropicalmed-08-00131-t005:** Resistance of *Anopheles minimus* to insecticides in Hainan.

Method	Year	Insecticide	Location	LC_50_ (mg a.i./L) or LT_50_ (min)or KT_50_ (min) (95% Confidence Interval) or Knockdown Rate after 1 h Exposure (%)	Toxicity Regression Line/ Mortality after 24 h Exposure	Resistance Index/ Resistance Level	Reference
WHO tube method	1978	DDT	Danzhou	LC_50_ = 5.5 mg/L	Y = 7.617 + 1.268X	-	[64]
	1981		Dongfang	LC_50_ = 4.8 mg/L	Y = 9.294 + 2.015X	-	[64]
WHO tube method at diagnosis dose	2006	0.025% deltamethrin	Changjiang	KT_50_ = 5.87 (3.68–9.39) min	Y = 3.8288 + 0.6618X	Sensitive	[6]
		0.01% cyfluthrin		KT_50_ = 25.33 (19.57–32.78) min	Y = 1.1173 + 1.2013X	Possible resistance	[6]
	2009	4% DDT	Dongfang	Knockdown rate = 96.3%	Mortality = 98.1%	Sensitive	[66]
		0.05% deltamethrin		Knockdown rate = 99.0%	Mortality = 99.0%	Sensitive	[66]
		0.15% cyfluthrin		Knockdown rate = 100%	Mortality = 100%	Sensitive	[66]
		5% malathion		-	Mortality = 100%	Sensitive	[66]

LC_50_, Lethal concentration 50% (the concentration expected to kill exactly 50% of exposed mosquitoes). LT_50_, Lethal time 50% (the time expected to kill exactly 50% of exposed mosquitoes at a certain dose). KT_50_, Knockdown time 50% (the time expected to knockdown exactly 50% of exposed mosquitoes at a certain dose).

## Data Availability

Data available on request from the authors.

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
