# Peer review of "Surveillance and Control of Malaria Vectors in Hainan Province, China from 1950 to 2021: A Retrospective Review"

_tropicalmed, 2023, doi:10.3390/tropicalmed8030131_

Round 1
Reviewer 1 Report
An article written by Sun et all entitled Research on malaria vectors from malaria epidemic to eliminatetion in Hainan Province, China (1950-2021), is a review of articles that are very interesting and informative, especially in contributing to the control of malaria vectors in the city of Huan China. Apart from that, this article also presents data on the trend of malaria disease and the existence of its vector from 1950 to 2021. I agree with the sentence conveyed by the authors that, more malaria vector research should be updated to inform decision-making for malaria vector control strategies (see the summary references especially research blood preferences and seasonability studies are very limited).
However, to add to and improve this review article, there are a few comments from me, including:
1. According to the author, what kind of entomology intervention is most effective in handling malaria vectors in Huan City, China?
2. Summary of references presented, maybe it is better to change the presentation so that it is easier for the reader, is it make Barr, a diagram to show the number of references per year
3. During those years, was there no information at all about vector or malaria control in China, using a herbal medicine approach?
4. Writing in sub-chapters or titles with the Latin name Ano. (Anopheles) should be consistent with italics
Reviewer 2 Report
1. Title: The title doesn’t precisely reflect the nature of this study. I would suggest using a new title like “Surveillance and control of malaria vectors in Hainan Province, China from
1950 to 2021: a retrospective review”.
2. Abstract: The abstract lacks a broader background (see my detailed comments regarding the Introduction) and their major conclusions. Instead of simply listing the number of papers for each specific research field, authors should be able to further summarize what they can conclude after reviewing these papers. For example, what surveillance and control measures have been taken and how these measures might have contributed to the elimination of malaria vectors, and what’s knowledge gap for future studies.
3. Introduction: The impact of the introduction can be significantly improved by starting with a broader background, such as a general introduction of current malaria situation and their vectors in the world, why and how surveillance and control of malaria vector matters in general. Then start focusing on Hainan Province by introducing its natural conditions and socioeconomic characteristics (dwelling conditions, e.g., population density, bed nets, window screen) that allow malaria vectors to thrive and transmit the parasite. Finally identify the knowledge gaps/research questions, highlight the potential impact of your research questions, and list your research objectives/goals and paths to achieve your goals.
4. Methods: The reviewing and screening effort can be specified to each author and the inclusion criteria should be specified too. The entire process of searching, reviewing, and screening literatures should better be summarized by a diagram (e.g., PRISMA flow diagram). At the end of this section, authors should briefly state how they will present the major findings (what contents are included and why these contents, the order of contents) in their results section for clarity.
5. Results: As mentioned above, the results should be clearly presented following a fixed structure; Given the subsections are not clear in the section, I would suggest using number to specify the order of each subsection and its belonging subsections; If most of the results only focused on An. minimus and An. dirus, why there is a last subsection for An. sinensis? At the end of each subsection, there should be a couple of sentences briefly summarizing of the findings.
6. Discussion: The discussion is meant to be a place where the findings are analyzed in the context of the published literature. However, a good number of the paragraphs in the discussion lack references, which means that they are interpreting the results instead of discussing the findings relative what is already known. Many of these sentences should be weaved into the results so that the readers understand their results. This part should be intensively reorganized.
7. Conclusions: As mentioned above, the conclusions did not include the major findings and significance of this review.
8. The writing of this manuscript needs significant improvement.
Round 2
Reviewer 2 Report
The manuscript has been significantly improved. There are still a few minor points that need to be addressed:
1. Methods: the criteria for including/excluding studies during literature screening at each stage (title screening, abstract screening, full text screening) should be clearly stated. For example, literature screening will begin with the removal of duplicated studies; studies that do not contain what information in the titles and abstracts will be excluded during title and abstract screening; further screening based on full text will exclude studies that do not have what information, data, etc. state what sort of studies will be included in your final list of review?
2. Figures and tables: authors should consider adding a couple of sentences to briefly explain the figures or tables, especially what is the take home message from such a figure/table.
3. Line 1717-1718: Blood preference is important in determining the vectorial capacity of mosquitoes. Greater anthropophily results in higher vectorial capacity.
Any references for these two statements? To the best of my knowledge, no evidence has suggested that blood feeding preference in mosquitoes can directly determine their vectorial capacity. Higher degree of anthropophily doesn’t result in higher vectorial capacity. Instead, blood feeding preference only alters host-vector-pathogen contact rate and hence the transmission dynamic of mosquito-borne diseases via its impact on R0 depending on whether a preferred/avoided host can be a competent pathogen reservoir and/or a biting mosquito can be a competent vector (Takken & Verhulst, Annu Rev Entomol 2013, 58: 433-453; Yan et al., Biol Rev 2021, 96:1367-1385). For a comprehensive understanding of blood feeding preference in mosquitoes, please refer to those two important reviews.
4. Grammar, spelling and typos need to be double-checked throughout the manuscript: for example, Line 2780 “… according to results our review” is missing an “of”.
